# Radioimmunotherapy Targeting IGF2R on Canine-Patient-Derived Osteosarcoma Tumors in Mice and Radiation Dosimetry in Canine and Pediatric Models

**DOI:** 10.3390/ph15010010

**Published:** 2021-12-22

**Authors:** Jaline Broqueza, Chandra B. Prabaharan, Kevin J. H. Allen, Rubin Jiao, Darrell R. Fisher, Ryan Dickinson, Valerie MacDonald-Dickinson, Maruti Uppalapati, Ekaterina Dadachova

**Affiliations:** 1College of Pharmacy and Nutrition, University of Saskatchewan, Saskatoon, SK S7N 5E5, Canada; mlb457@mail.usask.ca (J.B.); kja782@mail.usask.ca (K.J.H.A.); ruj501@mail.usask.ca (R.J.); 2Department of Pathology and Laboratory Medicine, College of Medicine, University of Saskatchewan, Saskatoon, SK S7N 5E5, Canada; chp347@mail.usask.ca; 3Versant Medical Physics and Radiation Safety, Kalamazoo, MI 49007, USA; darrell.fisher@versantphysics.com; 4Department of Veterinary Pathology, Western College of Veterinary Medicine, University of Saskatchewan, Saskatoon, SK S7N 5B4, Canada; ryan.dickinson@usask.ca; 5Department of Small Animal Clinical Sciences, Western College of Veterinary Medicine, University of Saskatchewan, Saskatoon, SK S7N 5B4, Canada; valerie.macdonald@usask.ca

**Keywords:** osteosarcoma, IGF2R, radioimmunotherapy, canine-patient-derived Gracie tumors, human dosimetry, canine dosimetry, ^177^Lutetium

## Abstract

Background: Osteosarcoma (OS) has an overall patient survival rate of ~70% with no significant improvements in the last two decades, and novel effective treatments are needed. OS in companion dogs is phenotypically close to human OS, which makes a comparative oncology approach to developing new treatments for OS very attractive. We have recently created a novel human antibody, IF3 to IGF2R, which binds to this receptor on both human and canine OS tumors. Here, we evaluated the efficacy and safety of radioimmunotherapy with ^177^Lu-labeled IF3 of mice bearing canine-patient-derived tumors and performed canine and human dosimetry calculations. Methods: Biodistribution and microSPECT/CT imaging with ^111^In-IF3 was performed in mice bearing canine OS Gracie tumors, and canine and human dosimetry calculations were performed based on these results. RIT of Gracie-tumor-bearing mice was completed with ^177^Lu-IF3. Results: Biodistribution and imaging showed a high uptake of ^111^In-IF3 in the tumor and spleen. Dosimetry identified the tumor, spleen and pancreas as the organs with the highest uptake. RIT was very effective in abrogating tumor growth in mice with some spleen-associated toxicity. Conclusions: These results demonstrate that RIT with ^177^Lu-IF3 targeting IGF2R on experimental canine OS tumors effectively decreases tumor growth. However, because of the limitations of murine models, careful evaluation of the possible toxicity of this treatment should be performed via nuclear imaging and image-based dosimetry in healthy dogs before clinical trials in companion dogs with OS can be attempted.

## 1. Introduction

Osteosarcoma (OS) is a primary malignant tumor of the bone, which makes it the fifth most common primary bone cancer among adolescents and young adults [1]. In spite of the significant efforts directed at finding new treatments, the overall patient survival rate is estimated at around 70%, with no significant improvement in survival achieved in the last two decades [2]. The prognoses for patients with metastases to the lungs and to the bone are particularly unfavorable, resulting in overall survival rates of 40% and 20%, respectively. The distinct feature of OS is its genetic variability between tumors from different patients [3]. Since the number of patients diagnosed every year with OS is not high, it is even more challenging to find a common feature in OS tumors that could be developed into a druggable target. Interestingly, canine OS in companion dogs, especially in large and giant breeds, demonstrates clinical presentations and molecular aberrations which are comparable to human patients with OS [4]. According to Morris Animal Foundation, every year, approximately 10,000 new canine OS cases are diagnosed in the US, with >90% of these dogs dying from systemic metastasis within 12 months of diagnosis.

As canine and human OS share certain antigens, this would allow for the potential targeting of such antigens with the same targeting molecules, e.g., antibodies [5,6], thus enabling a comparative oncology approach to finding effective treatments for human and canine OS. Cation-independent mannose 6-phosphate/insulin-like growth factor 2 receptor (IGF2R) has been shown to be overexpressed by the majority of commercial and patient-derived human OS cell lines [7]. Previously, a single-nucleotide polymorphism (SNP) within a haplotype block in IGF2R was found to correlate with an increased risk of OS development [8]. Our group has recently shown a consistent expression of IGF2R on the tumors from 34 consecutive canine OS patients treated for their condition at the Western College of Veterinary Medicine in Saskatoon, Canada [9]. Taken together, the consistent expression of IGF2R on the surface of human and canine tumors makes them an attractive druggable target for developing novel therapies for OS.

Targeted radionuclide therapy (TRT) involves the use of a radiopharmaceutical drug that specifically targets cancer cells [10]. We have previously shown the preferential tumor localization of radiolabeled murine monoclonal antibodies (mAbs) to human IGF2Rs (mAb MEM-238) and to human and murine IGF2Rs (mAb 2G11) in OS xenografts and patient-derived xenografts in mice in comparison with the control mAbs. The radioimmunotherapy (RIT) of OS tumors with IGF2R-specific mAbs MEM-238 and 2G11, which are 188Rhenium-labeled (188Re-labeled) and ^177^Lutetium-labeled (^177^Lu-labeled), respectively, resulted in tumor growth inhibition and even regression in some mice, with acceptable safety. Impressively, this effect of the RIT on the tumors in comparison with multiple control groups was achieved using a single administration and non-optimized dosing [11,12]. As modern requirements for biological drugs require the mAbs to be humanized or fully human, we have recently developed a fully human mAb IF3, which binds with high specificity and affinity to human, canine and murine IGF2R, and evaluated it in vitro and in vivo for binding to commercial OS cell lines and human and canine-patient-derived tumors [13]. In this work, we evaluated the efficacy and safety of radioimmunotherapy with the ^177^Lu-labeled IF3 of mice bearing canine-patient-derived tumors and performed canine and human dosimetry calculations.

## 2. Results

### 2.1. Biodistribution and Microspect/CT Imaging Demonstrated a High Uptake of the IF3 Antibody in Canine-Patient-Derived Gracie Tumors

Figure 1 shows the results of the biodistribution of ^111^In-IF3 in female SCID mice bearing canine-patient-derived Gracie tumors. The highest tumor uptake of approximately 18% ID/g was reached at 24 h post-injection. At all time points, the tumor to blood uptake ratio was >4. The IGF2R-expressing spleen uptake also reached its highest value of 50% ID/g at 24 h post-administration. The accumulations of ^111^In-IF3 in the pancreas, small intestines, bone and the spine reached their peak values at 48 h, with washout observed at 72 h. The metabolic organs’ (kidney, liver, stomach) uptake was unremarkable. The only organs which displayed a continuous uptake of the radiolabeled antibody were the lungs. The microSPECT/CT imaging of the tumor-bearing mice also demonstrated ^111^In-IF3 uptake in the tumor and the spleen (Figure 2).

### 2.2. Dosimetry Calculations Identified the Organs to Receive the Highest Radiation Dose in a Canine and Pediatric Patient

The radiation absorbed doses were calculated from the mouse biodistribution data and projected to the dog and child using methods consistent with the recommendations of the special committee on Medical Internal Radiation Dose (MIRD) of the Society of Nuclear Medicine and Medical Imaging (SNMMI). Table 1 displays the radiation doses which would be delivered by ^177^Lu-IF3 mAb to the OS tumor and major organs, calculated using the biodistribution data described above as applied to the model of a 32 kg 10-year-old female child and a 9.45 kg dog. The organs which would receive the highest radiation dose in the course of RIT of a 10-year-old female child would be, in descending order: spleen, tumor, skeletal surfaces, pancreas and lungs. In an average-sized dog, these organs, in descending order, would be: tumor, spleen, pancreas. The projected total body dose would be approximately two times higher in a pediatric patient than in a canine patient.

### 2.3. RIT with ^177^Lu-IF3 Was Highly Effective in Abrogating Canine-Patient-Derived-Gracie-Tumor Growth in SCID Mice

Figure 3A shows the results of the RIT experiment where tumor-bearing mice were treated with 60 µCi ^177^Lu-IF3 mAb with or without pre-blocking with unlabeled (“cold”) IF3 mAb. There was highly significant abrogation in tumor growth of RIT-treated mice in comparison with untreated controls (*p* = 0.0175) and with the “cold” IF3 group (*p* = 0.0025).

The RIT treatment was accompanied by weight loss in the mice treated with RIT, which was less pronounced in a group preblocked with “cold” IF3 before RIT administration (Figure 3B). This weight loss necessitated the sacrifice of the mice in the RIT group on day 13 after treatment initiation and the sacrifice of the mice in the RIT with the preblocking group on day 18. Analysis of the Kaplan–Meier survival curves (Figure 3C) demonstrated the statistically significant difference (*p* = 0.01) in survival between the RIT and RIT with preblocking groups. The spleens of the mice treated with RIT without pre-blocking with the unlabeled IF3 antibody were somewhat smaller than the spleens of the untreated mice, and some of the mice from the RIT group were showing petechiae on their ears (Figure 3D).

### 2.4. Immunohistochemistry of Canine Spleens Demonstrated Low Expression of IGF2R

The immunohistochemistry of spleens from two dogs, performed with IGF2R-specific murine mAb 2G11, demonstrated relatively low expressions of IGF2R in comparison with canine placenta used as a positive control (Figure 4). The staining was observed in the minority of cells, presumably macrophages, and mirrored that of the non-specific control murine mAb MOPC21.

## 3. Discussion

As part of our efforts to develop a treatment for OS using a comparative oncology approach, in this study, we evaluated the efficacy and safety of RIT with a novel human IGF2R-targeting IF3 antibody labeled with the theragnostic radionuclide ^177^Lu in mice bearing canine-patient-derived tumors, as well as performed dosimetry calculations in canine and human pediatric models. In this regard, using 3D cultures of canine OS cells or canine tumors in mice is a recognized way to gain therapeutic and mechanistic insights about proposed therapies before conducting clinical trials in companion animals [14,15].

The biodistribution of ^111^In-IF3 in Gracie-tumor-bearing mice and microSPECT/CT imaging demonstrated pronounced uptake in the tumors and fast clearance from the blood. The RIT experiments with ^177^Lu-IF3 which followed demonstrated the canine-patient-derived Gracie tumors were very responsive to a relatively low dose of 60 µCi ^177^Lu-IF3, which completely abrogated tumor growth. Using pharmacological recalculation factors which take into consideration the differences between body weight to body surface ratios in mice and humans [16], a 60 µCi dose would be equivalent to 14 mCi in a 70 kg adult or 7 mCi in a 35 kg child. Interestingly, the preblocking of IGF2R-binding sites by the administration of an approximately 10-fold higher amount of “cold” IF3 than that used in the radiolabeled formulation did not affect the efficacy of RIT while significantly prolonging the survival of the treated mice. This observation can most likely be explained by it blocking some of the IGF2Rs in the mouse spleens, resulting in the reduced uptake of ^177^Lu-IF3 in this organ and, as a result, less splenic toxicity. In human patients treated with ^177^Lu-DOTATE, splenic radiation dose correlated with the hematologic toxicity and the size of the spleens [17].

Radiation dosimetry calculations using models of a medium-sized dog and a 10-year-old female child showed that the dose to the tumor will be highest in a dog and second highest (after the spleen) in a child, which attests to the ability of the IF3 mAb to deliver a tumoricidal amount of its radioactive payload to the tumor. Interestingly, the dose to the skeletal surfaces in a child did not result in an appreciable dose to the bone marrow, which might be a consequence of the relatively short range of ^177^Lu beta emissions in tissues. Both models showed a high dose to the spleen, which is the consequence of the high uptake of the IF3 in a mouse spleen being extrapolated to the canine and human models. However, according to the Human Protein Atlas Database, IGF2Rs are only minimally or moderately expressed in all normal organs in humans except for the testes, suggesting that using it as a target for RIT should be relatively safe: https://www.proteinatlas.org/ENSG00000197081-IGF2R/tissue (accessed on 1 November 2021). For comparison, SSTR1, which is targeted by the clinically approved Lutathera, a ^177^Lu-labeled peptide, is highly expressed in several crucial organs and is still safe to administer to patients: https://www.proteinatlas.org/ENSG00000139874-SSTR1/tissue (accessed on 1 November 2021). The immunohistochemistry of two canine spleens revealed only a low uptake of IGF2R by the minority of cells in the spleens. Future PET or SPECT imaging of dogs with radiolabeled IF3 should clarify the question of spleen uptake for canine patients. Overall, the dosimetry results demonstrated that the doses which would be delivered to all major organs, including the spleen, during RIT with ^177^Lu-IF3 would be below the doses with Lutathera [17,18] and below the maximum tolerated doses for major organs defined in [19].

This study has several limitations. Firstly, our SCID mouse model has an IGF2R-specific limitation of very high expression of this protein by the spleen, which served as a “sink” of the radiolabeled antibody and contributed to the toxicity of the treatment. A second limitation, which is generally related to mouse models, is that murine FcRn receptors have a higher affinity for Fc fragments of human antibodies than for Fc fragments of murine antibodies, thereby leading to a shorter half-life [20]. For RIT applications, the binding of the radiolabeled antibodies to Fc receptors in the bone marrow, spleen, etc., might lead to hematologic toxicity. The third limitation is related to dosimetry. The extrapolation from a mouse to another species requires a very high number of individual calculations, and the direct extrapolation between species based on their ratios of organ mass is tentative and not fully representative of the potential differences in species’ metabolic rates.

## 4. Materials and Methods

### 4.1. Antibody, Radionuclides and Radiolabeling

Human, canine and murine IGF2R-binding IF3 mAb was expressed and purified in our laboratories as described in [13]. SG-iTLC (silica gel instant thin layer chromatography) strips for quantification of radiolabeling yields were acquired from Agilent (Mississauga, ON, Canada). (R)-2-Amino-3-(4-isothiocyanatophenyl)propyl]-trans-(S,S)-cyclohexane-1,2-diamine-pentaacetic acid (CHXA’’) bifunctional chelating agent was purchased from Macrocyclics (Plano, TX, USA). ^111^In was obtained from BWXT (Cambridge, ON, Canada), and ^177^Lu, from RadioMedix (Houston, TX, USA). IF3 mAb was conjugated to 2.5 initial molar excess of CHXA” as in [13]. The conjugate–antibody ratio (CAR) of CHXA” molecules per IF3 molecule post-conjugation was determined by MALDI-TOF (University of Alberta, Edmonton, Canada) to be 0.91. Radiolabeling with ^111^In and ^177^Lu was performed as in [13]. Radiolabeling yields were typically greater than 98%, and radiolabeled mAbs required no further purification.

### 4.2. Animal Models

Healthy six- to eight-week-old SCID (CB17/Icr-Prkdcscid/IcrIcoCrl) female mice obtained from Charles River Laboratories (Wilmington, MA, USA) were used for the biodistribution and therapy experiments. The canine-patient-derived osteosarcoma Gracie cell line was a kind gift from Dr. Doug Thamm’s lab at Colorado State University School of Veterinary Medicine, USA. Gracie cells were grown as in [13]. Mice were anesthetized with isoflurane for tumor induction and were injected with 4 × 10^6^ Gracie cells in their right flanks. Mice were monitored thrice per week for tumor development.

### 4.3. Biodistribution and MicroSPECT/CT of ^111^In-Labeled IF3 in Tumor-Bearing Mice

Mice bearing Gracie tumors were randomized into groups of five and were injected intraperitoneally with 18 µCi of ^111^In-IF3. At 2, 24, 48 and 72 h post-injection, mice were sacrificed and the following organs were collected: blood, tumor, heart, lungs, pancreas, spleen, kidney, liver, brain, stomach, small intestine, large intestine, thigh muscle and bone. The percentage injected dose per gram (%ID/g) was then calculated by weighing the organ and counting the radioactivity with a gamma counter (PerkinElmer, Waltham, MA, USA). MicroSPECT/CT (micro single-photon emission computer tomography/computer tomography) images were collected on a MILabs VECTor^4^ (Houten, The Netherlands) microSPECT/CT at 24 and 48 h post ^111^In-IF3 administration (200 µCi via intravenous injections) and processed as in [13].

### 4.4. Dosimetry Calculations

Radiation doses for human and canine subjects were extrapolated and calculated from the mouse data. MIRD formalism, implemented using OLINDA v2.0, and other direct principles, were used for dosimetry calculations. The average percent administered activity per gram (%IA/g) in the mouse was obtained and group-averaged for each time point and each organ or tissue. We used the relative-organ-mass scaling method of Molina-Trinidad et al. [21] to calculate an extrapolated set of values, repeated at each time point, for the organ masses given in the 9.45 kg canine and the 32 kg pediatric models, as follows:%Am g,organhuman  =%Am g,organmouse ∗ m kg,body mouse ∗ m g,organm kg,bodyhuman
%Am g,organcanine  =%Am g,organmouse ∗ m kg,body mouse ∗ m g,organm kg,bodycanine
where A is the activity administered, measured at each time point, and m is the organ mass (grams) or whole-body mass (kg).

The above scaling formula assumes that the metabolic retention and clearance from the mouse can be extrapolated to a human based on the mass ratios given above. This method has been widely used and cited by others; however, direct extrapolation between species based on ratios of organ mass remains tentative and perhaps not fully representative of potential differences in species’ metabolic rates. Using this method, new datapoints for (1) the 9.45 kg model and (2) the 32 kg female pediatric model were obtained. The new datapoints for each phantom model were then fitted by least-squares regression analysis to a preferred function and the function was integrated to infinity. The time-integrated activity coefficients for each organ or tissue were entered into OLINDA [22] for the dog model and for the pediatric human model. Radiation doses for ^177^Luwere obtained in units of either mSv/MBq organ dose equivalent or centigray (cGy) absorbed dose per millicurie administered. Differences in calculated results occur because of differences in the phantom models. Model assumptions are variable between the two phantoms for heart, heart wall, small intestine and large intestine (such as right and left colon). Radiation doses for muscle and blood were calculated from first principles using the time-integrated activity coefficients for ^177^Lu in each, since neither of the phantom models employs blood or muscle as source–target organ pairs.

### 4.5. Therapy of ^177^Lu-Labeled IF3 in Tumor-Bearing Mice

Mice were monitored and randomized into groups of five once tumor size reached ~50–100 mm^3^. The therapy study included four groups: group 1 was untreated; group 2 received unlabeled IF3 (cold); group 3 received 60 μCi of ^177^Lu-IF3; and group 4 received 60 μCi of ^177^Lu-IF3 with pre-blocking. Mice in group 4 received 200 µg of the IF3 antibody 2 h before treatment. Tumors and mouse body weights were then monitored three times per week. The formula V = (L × W × W)/2 was used to calculate the tumor volume for each mouse. Mice were humanely sacrificed when they reached the Humane Intervention Point (HIP) described in our animal use protocol 20170006, namely, the animals were humanely euthanized if they experienced excessive weight loss (≤20%), became moribund or any tumor reached 4000 mm^3^ volume or became necrotic.

### 4.6. Immunohistochemistry

The IGF2R-specific mAb 2G11 and the isotype-matching control mAb MOPC21 were obtained from Thermo Fisher (Saskatoon, SK, Canada). Immunohistochemistry of the spleens from two dogs was performed as described in [12], with the only difference being using 1:100 2G11 mAb dilution. Canine placenta was used as a positive control for IGF2R by analogy with a human placenta [23].

### 4.7. Statistical Analysis

Power analysis for the in vivo studies was estimated using PASS version 11 (NCSS, Inc., Kaysville, UT, USA), using simulations of different tumor volumes based on pilot data and conservative assumptions regarding the groups treated with the radiolabeled antibodies. All simulations showed power of at least 83% with only five animals per group because of the large differences between treated and untreated animals. Thus, five mice per group were utilized in the in vivo studies. GraphPad Prism 7 and Microsoft Excel were used to analyze all the data (GraphPad Software, Inc., San Diego, CA, USA). Differences between the treated and untreated groups in vivo were assessed using nonparametric Kruskal–Wallis test with Dunn’s correction for multiple comparisons. Error bars represent ±standard deviation (SD). Kaplan–Meier data were analyzed by log-rank (Mantel–Cox) test.

## 5. Conclusions

In conclusion, these results demonstrate that RIT with ^177^Lu-IF3 targeting IGF2Rs on experimental canine OS tumors effectively decreases tumor growth. However, due to the limitations of murine models, a careful evaluation of the possible toxicity of this treatment should be performed via nuclear imaging and image-based dosimetry in healthy dogs before clinical trials in companion dogs with OS can be attempted.

## Figures and Tables

**Figure 1 pharmaceuticals-15-00010-f001:**
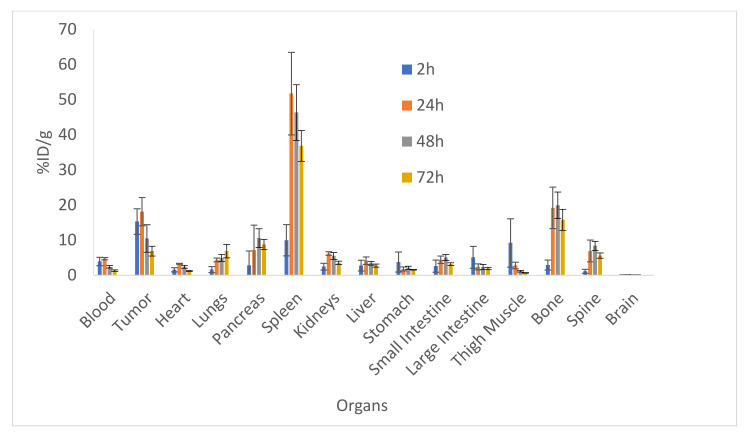
Biodistribution of ^111^In-IF3 mAb to IGF2R in female SCID mice bearing canine-patient-derived Gracie tumors after IP administration at 2, 24, 48 and 72 h post-administration.

**Figure 2 pharmaceuticals-15-00010-f002:**
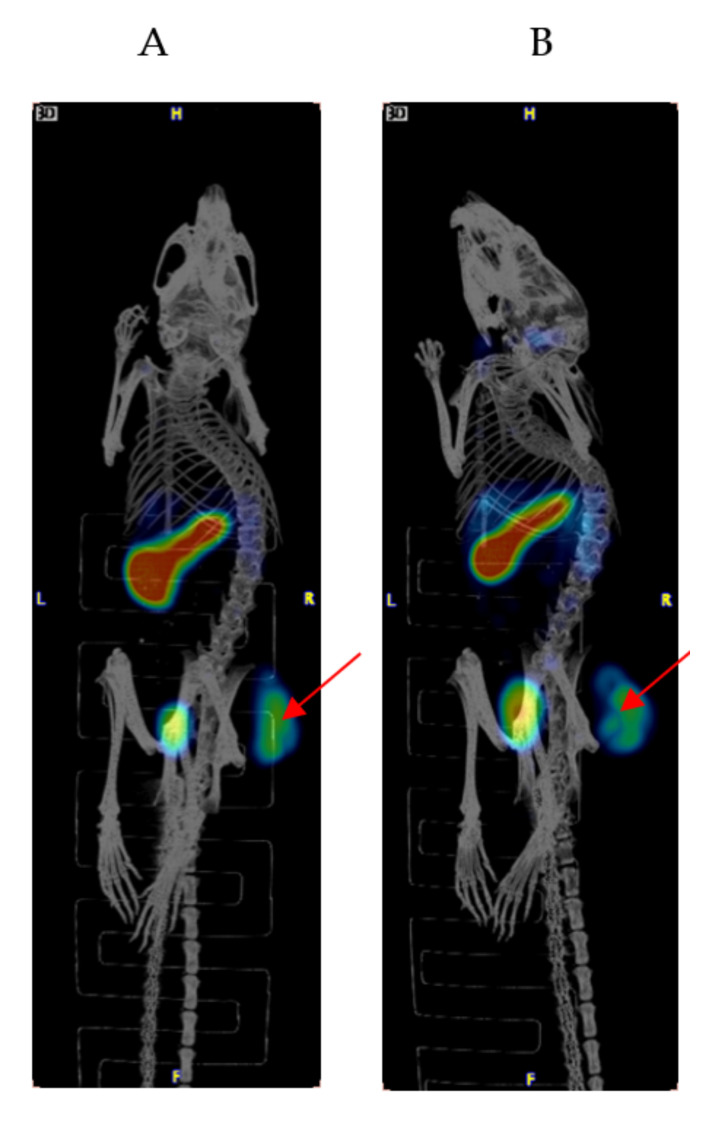
MicroSPECT/CT imaging of a female SCID mouse bearing canine-patient-derived Gracie tumors injected via IV with ^111^In-IF3 mAb to IGF2R. (**A**) 24 h post-injection; (**B**) 48 h post-injection. The uptake in the tumors and spleen and excretion through the bladder are seen. Red arrows are pointing to the tumors.

**Figure 3 pharmaceuticals-15-00010-f003:**
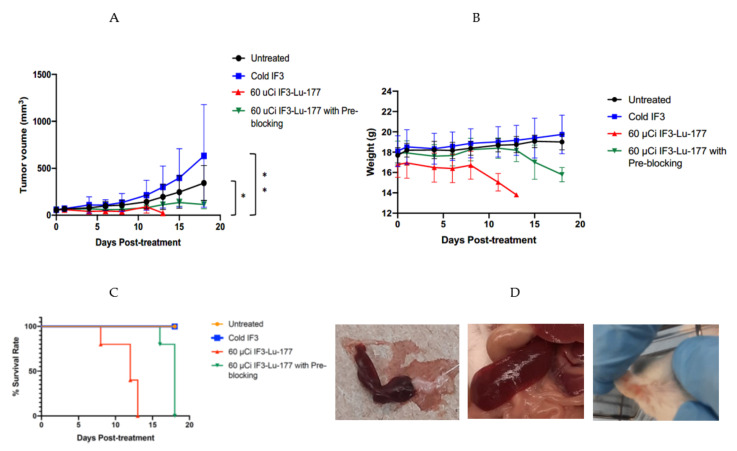
RIT of SCID mice bearing Gracie tumors with ^177^Lu-IF3 mAb: (**A**) tumor volume; (**B**) mouse body weight; (**C**) Kaplan–Meier survival curves. Groups of 5 mice were treated with either: 60 µCi ^177^Lu-IF3 mAb; or 60 µCi ^177^Lu-IF3 mAb preceded by 200 µg of “cold” IF3 antibody 2 h before RIT; or 12 µg “cold” IF3 mAb; or left untreated; (**D**) gross pathology of the mice post-mortem: left plate—spleen of RIT-alone-treated mouse; middle plate—spleen of untreated mouse; right plate—petechiae on the ears of RIT-alone-treated mouse. * means *p* = 0.01, ** means *p* = 0.001.

**Figure 4 pharmaceuticals-15-00010-f004:**
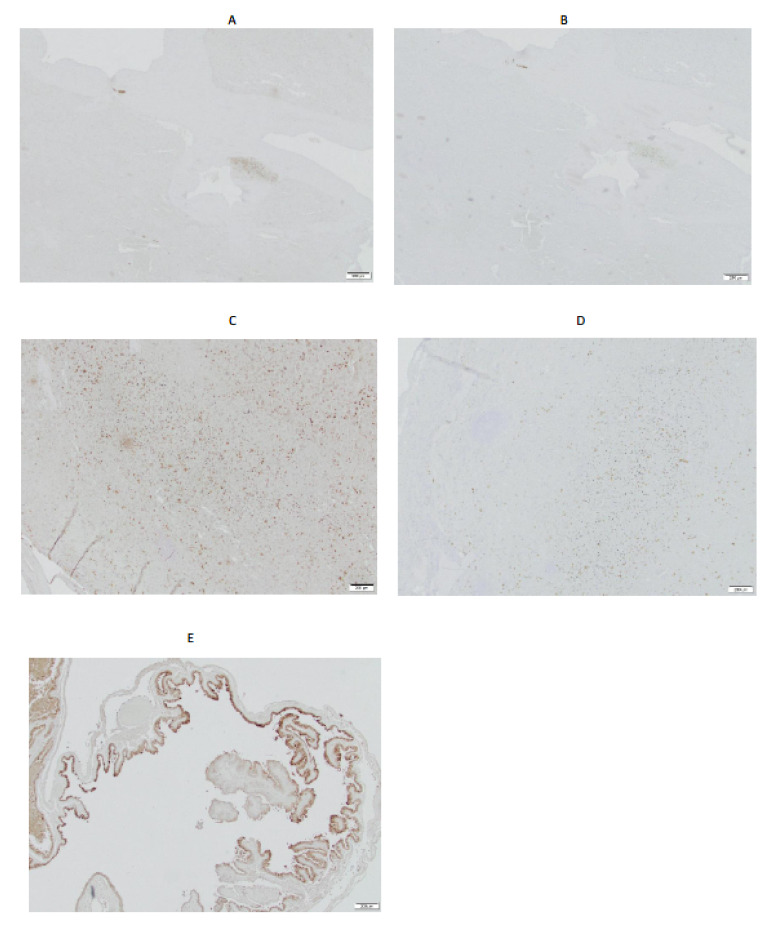
Immunohistochemistry of canine spleens with 2G11 IGF2R-specific antibody. (**A**) spleen, case # PDS 2114514, 2G11 mAb; (**B**) spleen, case # PDS 2114514, MOPC21 control mAb; (**C**) spleen, case # 2128371, 2G11 mAb; (**D**) spleen, case # 2128371, MOPC21 control mAb; (**E**) canine placenta, 2G11 mAb. Size bar—200 μm.

**Table 1 pharmaceuticals-15-00010-t001:** Radiation doses which would be delivered by ^177^Lu-IF3 mAb to the OS tumor and major organs in a 32 kg 10-year-old female child and a 9.45 kg dog.

	32 kg Human Model		9.45 kg Dog Model	
Target Organ	mSv/MBq	cGy/mCi	mSv/MBq	cGy/mCi
Adrenals	0.0434	0.161	--	--
Brain	0.0107	0.0396	0.0740	0.274
Heart Wall	0.0469	0.174	0.166	0.614
Eyes	0.00166	0.00614	0.0204	0.755
Gallbladder Wall	0.00376	0.0139	0.0454	0.168
Small Intestine	0.0640	0.237	0.0753	0.279
Stomach Wall	0.0116	0.0429	0.0848	0.314
Large Intestine	--	--	0.120	0.444
Right colon	0.0695	0.257	--	--
Rectum	0.00116	0.00429	--	--
Kidneys	0.0886	0.328	0.0885	0.327
Liver	0.0648	0.240	0.0638	0.236
Lungs	0.397	1.47	0.0577	0.213
Pancreas	0.509	1.88	0.0792	0.293
Tumor	0.785	2.90	0.377	1.39
Red Marrow	0.00300	0.0111	--	--
Skeletal Surfaces	0.607	2.25	0.0584	0.216
Bone spine	0.211	0.781	0.0185	0.0685
Spleen	3.06	11.3	0.230	0.851
Thymus	0.00752	0.0203	0.0242	0.0895
Thyroid	0.00610	0.0226	0.00410	0.0152
Urinary Bladder	0.000838	0.00310	0.0421	0.156
Blood	0.0210	0.0777	0.0210	0.0777
Muscle	0.0790	0.292	0.0790	0.292
Total Body	0.0496	0.183	0.0256	0.0947

## Data Availability

All data presented in this study are available in this article.

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
