# Peer review of "Radioimmunotherapy Targeting IGF2R on Canine-Patient-Derived Osteosarcoma Tumors in Mice and Radiation Dosimetry in Canine and Pediatric Models"

_pharmaceuticals, 2021, doi:10.3390/ph15010010_

Round 1
Reviewer 1 Report
This is a murine study that shows potential of anti-IGFR radiopharmaceuticals for bone sarcoma treatment (osteosarcoma and perhaps also Ewing sarcoma). A pre-treatment with cold antibody may improve targeting and this will be important in future study in larger animals. Figure 3C seems to be missing both lines for untreated and for blue cold IF3 (would expect shorter survival if survival was significantly better as indicated in the text). I look forward to canine studies on companion dogs with osteosarcoma which should confirm calculations - and potential for clinical safety and efficacy in this paper.
Author Response
This is a murine study that shows potential of anti-IGFR radiopharmaceuticals for bone sarcoma treatment (osteosarcoma and perhaps also Ewing sarcoma). A pre-treatment with cold antibody may improve targeting and this will be important in future study in larger animals. Figure 3C seems to be missing both lines for untreated and for blue cold IF3 (would expect shorter survival if survival was significantly better as indicated in the text). I look forward to canine studies on companion dogs with osteosarcoma which should confirm calculations - and potential for clinical safety and efficacy in this paper. – Response: We would like to thank the Reviewer for his/her encouraging opinion about our study. We apologize for the bad choice of colors for Fig. 3C which made several lines invisible. We have changed the colors in revised Fig.3C to make all the lines easily visible.

Reviewer 2 Report
The paper showed data relating to radioimmunotherapy targeting IGF2R of canine patient derived osteosarcoma (OS) tumors inoculated SQ into SCID mice. They reported calculations designed to estimate dosimetry in canine and pediatric models, but failed to fully describe the calculations that were performed or document their validity in any way. The dose chosen to treat mice bearing canine OS caused significant toxicity that was attributed to “splenic toxicity” (Line 258). No correlative hematologic or histologic data were provided to support the conclusion that death was due to “splenic toxicity”. Weight loss was the only data shown that was associated with decreased survival. An adequate description of criteria for euthanasia was not provided.
Data regarding the expression of IGF2R in canine splenic tissue was incomplete. Evidence of validation of the antibody used (mAb 2G11) was not provided for canine tissue. There was no discussion of potential off-target expression in the placenta of dogs or explanation of why canine placenta was chosen as a positive control tissue.
The authors’ conclusion that “results demonstrate that RIT with 177Lu-IF3 targeting IGF2R on canine OS tumors is effective” is alarmingly overreaching and is not supported by the data provided in this manuscript.
Author Response
The paper showed data relating to radioimmunotherapy targeting IGF2R of canine patient derived osteosarcoma (OS) tumors inoculated SQ into SCID mice. They reported calculations designed to estimate dosimetry in canine and pediatric models, but failed to fully describe the calculations that were performed or document their validity in any way. – Response: We apologize for the lack of details in describing dosimetric methodology. We have significantly expanded the dosimetry section in revised manuscript to clarify methodology used.
The dose chosen to treat mice bearing canine OS caused significant toxicity that was attributed to “splenic toxicity” (Line 258). No correlative hematologic or histologic data were provided to support the conclusion that death was due to “splenic toxicity”. Weight loss was the only data shown that was associated with decreased survival. An adequate description of criteria for euthanasia was not provided. – Response: We did not sample the blood of the mice in therapy study as these tumor-bearing severely immunocompromised mice were quite fragile during the study and we did not want to stress them any further. We have now included the criteria for euthanasia into the revised part of the Materials and Methods. The spleens of the mice treated with RIT without pre-blocking with the unlabeled IF3 antibody were somewhat smaller than the spleens of the untreated mice, and some of the mice from RIT group were showing petechiae. We have included this information into the Results and the respective images into the revised Fig. 3.
Data regarding the expression of IGF2R in canine splenic tissue was incomplete. Evidence of validation of the antibody used (mAb 2G11) was not provided for canine tissue. There was no discussion of potential off-target expression in the placenta of dogs or explanation of why canine placenta was chosen as a positive control tissue. – Response: It is known that human placenta expresses high levels of IGF2R which functions there both as a signaling and clearance receptor (Harris LK et al. Biol Reprod. 2011 Mar; 84(3):440-6; doi: 10.1095/biolreprod.110.088195). This is why for our initial study probing osteosarcoma tumors from two companion dogs with 2G11 mAb we used canine placenta which proved also to be strongly positive for IGF2R (ref. 12). We apologize for not including the reference to the human placenta into the submitted version, it is now included into the revised manuscript as new reference 23.
The authors’ conclusion that “results demonstrate that RIT with 177Lu-IF3 targeting IGF2R on canine OS tumors is effective” is alarmingly overreaching and is not supported by the data provided in this manuscript. – Response: We have completely reworded the conclusion which now states: In conclusion, these results demonstrate that RIT with 177Lu-IF3 targeting IGF2R on experimental canine OS tumors effectively decreases tumor growth. However, due to limitations of murine models, careful evaluation of possible toxicity of this treatment should be performed via nuclear imaging and image based dosimetry in healthy dogs before clinical trials in companion dogs with OS could be attempted.